# Virtual-Eyes: Quantitative Validation of a Lung CT Quality-Control Pipeline for Foundation-Model Cancer Risk Prediction

**Md. Enamul Hoq**[1]                                                        MHOQ@UAMS.EDU
**Linda Larson-Prior**[2]                                           LLARSONPRIOR@UAMS.EDU
**Fred Prior**[1]                                                         FWPRIOR@UAMS.EDU

[1] *Department of Biomedical Informatics, College of Medicine, University of Arkansas for Medical Sciences, Little Rock, AR, USA*

[2] *Department of Neuroscience, College of Medicine, University of Arkansas for Medical Sciences, Little Rock, AR, USA*

**Editors:** Accepted for publication at MIDL 2026

## Abstract

Robust preprocessing is rarely quantified in deep-learning pipelines for low-dose CT (LDCT) lung cancer screening. We develop and validate Virtual-Eyes, a clinically motivated, 16-bit CT quality-control pipeline for NLST, and measure its differential impact on generalist foundation models versus specialist models. Virtual-Eyes enforces strict $512 \times 512$ resolution, rejects short or non-diagnostic series, and extracts a contiguous lung block using Hounsfield-unit filtering and bilateral lung-coverage scoring while preserving the original 16-bit DICOM grid. Using 765 NLST patients (182 cancer, 583 non-cancer), we evaluate RAD-DINO, Merlin, Sybil, and ResNet-18 under a leakage-free protocol. For RAD-DINO, preprocessing improves slice-level AUC from 0.576 to 0.610 and patient-level AUC from 0.646 to 0.683 (mean pooling) and 0.619 to 0.735 (max pooling), with improved calibration (Brier score $0.188 \rightarrow 0.112$). In contrast, Sybil and ResNet-18 degrade under Virtual-Eyes, revealing reliance on contextual or shortcut features, while Merlin shows limited transferability. Sensitivity analysis and uncertainty estimation confirm the robustness and stability of these findings.

**Keywords:** Lung Cancer Screening, Foundation Models, Quality Control, CT Preprocessing, Validation

## 1. Introduction

Low-dose CT (LDCT) screening reduces lung-cancer mortality in high-risk populations (Aberle et al., 2011; Kramer et al., 2011; de Koning et al., 2020). As LDCT adoption grows, deep-learning pipelines increasingly rely on large-scale pretraining through medical foundation models (FMs), such as CT-FM (Pai et al., 2025), M3FM (Niu et al., 2025), RAD-DINO (Pérez-García et al., 2025; Hoq et al., 2025a), Merlin (Blankemeier et al., 2024), and other generalist encoders, to leverage unlabeled data across modalities and anatomies. However, most work focuses on model architecture and supervision, treating preprocessing as a fixed, often under-documented step.

Specialist models such as Sybil (Mikhael et al., 2023) and conventional backbones like ResNet-18 (He et al., 2016) are typically trained end-to-end on raw or minimally processed LDCT volumes and learn some robustness to scanner tables, cables, air regions, and

extra-thoracic anatomy. Many recent FMs, including M3FM and CT-FM, rely on auxiliary segmentation models or heuristic slice filters to define the lung field before pretraining or fine-tuning. These segmentation-based pipelines are computationally intensive, can introduce false positives and false negatives in the selected regions, and are often opaque in published work, making it difficult to disentangle the contribution of preprocessing from that of the model itself.

Generalist FMs, in contrast to specialist models, are trained on heterogeneous modalities and anatomies and may treat all visual context as equally informative. We therefore hypothesize that anatomically focused preprocessing is *necessary* to align LDCT with generalist FMs, but the same preprocessing can *harm* specialist models that adapted to raw NLST statistics and may partially rely on contextual shortcuts. Also, the contextual cues echoes earlier findings from real-time image correlation research, where background structure and acquisition geometry were shown to dominate learned correlations unless the visual domain was carefully constrained (Hoq, 2020), and aligns with recent multimodal pathology studies demonstrating that uncontrolled visual context can mislead both CNNs and vision–language foundation models during retrieval and caption generation (Hoq et al., 2025b).

To test this hypothesis, we develop *Virtual-Eyes*, a lung-aware 16-bit quality-control pipeline, and quantify its impact across four architectures: two generalist FMs (RAD-DINO, Merlin) and two specialist models (Sybil, ResNet-18). Unlike segmentation-heavy approaches, Virtual-Eyes is a deterministic CPU-based pipeline that operates directly on Hounsfield units, is free from false-positive/false-negative mask predictions, and can be reproduced on commodity hardware. We ask when such a simple, lung-focused QC step helps or hurts downstream cancer-risk prediction.

## 2. Methods

### 2.1. Dataset and Splits

We used a curated subset of the NLST LDCT cohort, accessed via The Cancer Imaging Archive (TCIA) (Clark et al., 2013; Aberle et al., 2011; Kramer et al., 2011; de Koning et al., 2020), comprising 765 unique patients (182 cancer-positive and 583 non-cancer). Each patient contributes all available baseline LDCT series. To avoid information leakage, we performed a strict patient-level split into training, validation, and test sets such that no patient appears in more than one partition. The final split consisted of 459 patients for training (109 cancer, 350 non-cancer), 153 for validation (37 cancer, 116 non-cancer), and 153 for held-out testing (36 cancer, 117 non-cancer). All models and preprocessing ablations were tuned on the validation set only, and all performance numbers reported in this work are computed on the fixed test set.

The NLST cohort includes multi-center LDCT acquisitions with variability in slice thickness, reconstruction kernels, and scanner vendors. This heterogeneity motivates the need for robust preprocessing and may influence both lung detection and downstream model performance.

## 2.2. Virtual-Eyes Quality-Control Pipeline

Virtual-Eyes is a deterministic, rule-based quality-control pipeline that operates directly on 16-bit DICOM CT series and enforces both scanner-level QC and lung-aware trimming before any learning-based model is applied. The implementation is written in Python using `pydicom`, `scikit-image`, and `torch`, and can run entirely on CPU; when available, Apple's Metal Performance Shaders (MPS) are used to accelerate voxel-wise operations but do not change the algorithmic behavior.

For each patient, all axial CT series are discovered by walking the DICOM directory tree and grouping files by `SeriesInstanceUID`. Within a series, slices are sorted along the superior–inferior axis using the `ImagePositionPatient` $z$-coordinate. Raw pixel arrays are converted to physical Hounsfield Units (HU) using the DICOM `RescaleSlope` and `RescaleIntercept` tags and stored as 32-bit floats. Series that fail basic scanner-level QC are rejected early: we discard any series with fewer than 64 axial images or with a matrix size other than $512 \times 512$. The minimum-length filter mirrors recent foundation-model pipelines for lung cancer screening such as M3FM, which likewise exclude very short or incomplete volumes during pretraining and evaluation (Niu et al., 2025). This step removes scout views, limited-range follow-up scans, and truncated series that would otherwise introduce highly heterogeneous context.

Lung-aware trimming is then performed slice-by-slice using a CPU lung-detection operator. For each axial slice, we first threshold voxels on MPS (or CPU fallback) to the canonical parenchymal range $[-950, -700]$ HU, which is widely used for lung densitometry and lung segmentation in clinical and research settings (Hofmanninger et al., 2020). The resulting binary mask is transferred to CPU and refined using `scikit-image` morphology: a disk-shaped opening (radius 2) suppresses isolated noise, followed by a closing operation (radius 5) to fill small gaps, consistent with established morphological post-processing techniques in automated lung segmentation (Armato and Sensakovic, 2004). Connected components are labelled, and only regions whose area exceeds 1% of the full $512 \times 512$ field-of-view are retained. The union of these regions defines a cleaned lung mask; if no such region is present, the slice is considered lung-free. For each slice we also compute a lung area ratio, defined as the fraction of in-plane pixels belonging to the cleaned mask. In practice, this ratio must exceed roughly 5% before a slice is consistently flagged as lung-containing, closely matching the configuration in our code (`MIN_LUNG_VOLUME_RATIO = 0.05`), and is consistent with prior knowledge-based lung-segmentation approaches that apply minimum region-area thresholds to distinguish true lung parenchyma from noise or extraneous anatomy (Brown et al., 2000).

Across the ordered slice stack of a series, Virtual-Eyes then converts the per-slice lung flags into a one-dimensional binary sequence and identifies all contiguous runs of lung-containing slices. Among these candidates, the algorithm selects the longest contiguous block and records its start and end indices. Blocks shorter than 20 slices are considered anatomically implausible for full-lung coverage and lead to series rejection (`MIN_BLOCK_SIZE = 20`). For accepted series, only this dominant lung block is retained; superior neck slices and inferior abdominal slices are permanently removed. This head-to-tail trimming logic is implemented purely through HU-based rules and connected-component geometry and does not rely on any learned segmentation network, thereby avoiding model-induced false-

positive or false-negative slice classifications. Although several thresholds in Virtual-Eyes (such as the HU window and minimum lung area ratio) are guided by prior work, other hyperparameters, including the precise choice of minimum contiguous block length and neighborhood margins, do not have strong support from the literature. For these, we systematically swept candidate values on NLST, visually inspected hundreds of scans across scanners and slice thicknesses, and selected the smallest thresholds that reliably preserved full lung coverage while removing neck and abdominal content. We acknowledge that these empirically tuned parameters are dataset- and protocol-dependent and will need to be re-validated and potentially adjusted when Virtual-Eyes is deployed on newer cohorts or institutions.

For each accepted series, Virtual-Eyes additionally performs lightweight intensity normalization. HU values within the retained lung block are clipped to a standard lung window (center $-500$ HU, width 1500) and linearly rescaled for visualization, but the data are saved to disk as 16-bit integers in the native $512 \times 512$ grid to preserve the original clinical dynamic range. The final per-series output is a single NumPy volume (`lung_block.npy`) containing only the contiguous lung block, stored under a directory hierarchy that encodes both the patient identifier and `SeriesInstanceUID`. This structure enables traceability back to the original DICOMs and supports downstream slice-level or volume-level embedding extraction. Additional qualitative examples of Virtual-Eyes trimming and accepted/rejected series are shown in Appendix A. Figure 1 depicts the complete pipeline of our work including preprocessing to validation.

After all series have been processed, Virtual-Eyes aggregates QC statistics into a CSV report. For each series, the report records the patient identifier, series UID, QC status (accepted/rejected), rejection reason, original number of slices, and number of slices kept. From this, we compute the total number of raw images, the number of images retained in lung blocks, and the proportion of slices discarded by QC. These summary statistics are used in the Results section to quantify how aggressively Virtual-Eyes reduces non-lung content before embedding extraction.

## 2.3. Hyperparameter Sensitivity Analysis

To assess robustness, we evaluated Virtual-Eyes under modest variations of key parameters, including the lung-area ratio threshold and minimum contiguous lung block size. Across these variations, RAD-DINO patient-level AUC remained stable, with only minor fluctuations, indicating that performance gains are not sensitive to precise parameter tuning. These results support the reliability of the chosen default configuration while highlighting the flexibility of the pipeline across acquisition variability.

## 2.4. Model Architectures, Embedding Pipelines, and Evaluation

We evaluated four complementary architectures spanning generalist and specialist regimes. RAD-DINO (Pérez-García et al., 2025) and Merlin (Blankemeier et al., 2024) are treated as frozen generalist encoders: after Virtual-Eyes, each lung slice is resized and normalized according to the model's requirements, passed through the encoder, and its latent representation is stored as a 1D embedding. Sybil is used as a specialist volumetric risk model

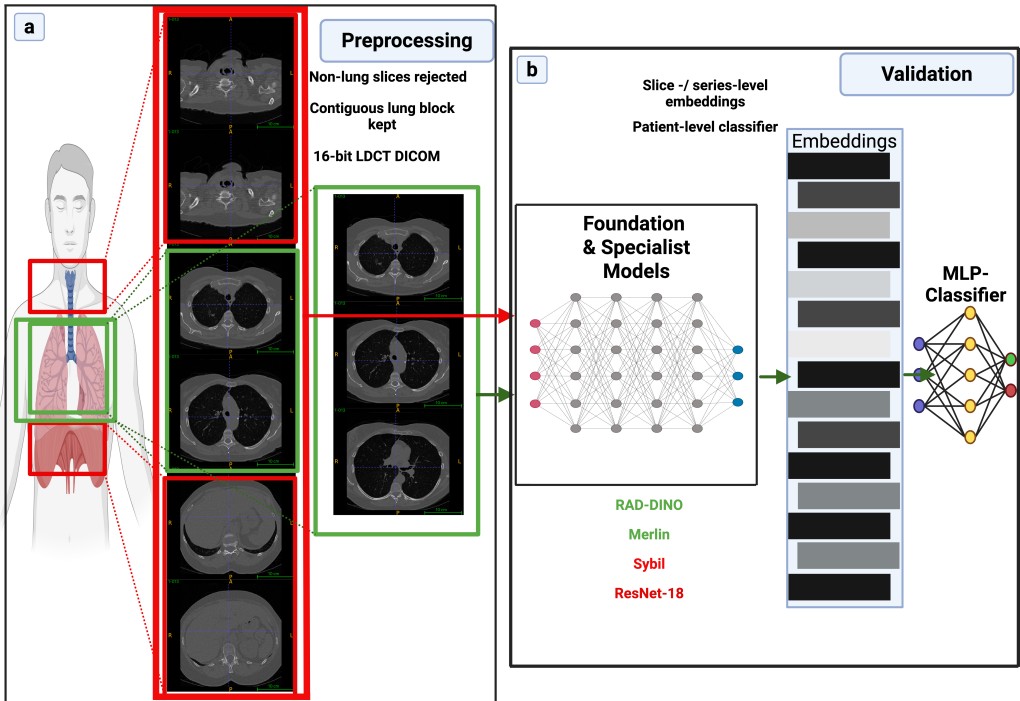

Figure 1: Overall Virtual-Eyes workflow. (**a**) Lung-focused preprocessing that rejects non-lung slices and extracts a contiguous 16-bit LDCT lung block. (**b**) Foundation and specialist models (RAD-DINO, Merlin, Sybil, ResNet-18) used to compute slice-/series-level embeddings, which are pooled into patient-level representations and fed to an MLP classifier for lung cancer risk prediction.

for LDCT (Mikhael et al., 2023), and a 2D ResNet-18 (He et al., 2016) serves as a more conventional convolutional baseline.

For RAD-DINO and Merlin, we train lightweight multi-layer perceptron (MLP) heads on top of slice embeddings. Each MLP consists of two fully connected layers with ReLU activation, dropout regularization, and a final sigmoid output, optimized using AdamW with a learning rate of $10^{-4}$ and mini-batches of 128 slices for 40 epochs. Training and early stopping are guided by validation ROC–AUC. Slice-level malignancy probabilities are then aggregated into a single patient-level risk score using several pooling strategies: mean pooling, max pooling, and top-$K$ pooling (with $K \in \{3, 5\}$ selecting the highest-risk slices). Sybil is evaluated by feeding either raw or Virtual-Eyes-preprocessed volumes into the published implementation without retraining; its native volumetric output is taken as the patient-level risk score. ResNet-18 is fine-tuned end-to-end on axial slices with binary cross-entropy loss, and patient-level scores are obtained via the same pooling strategies used for RAD-DINO and Merlin.

For all models, we compute receiver operating characteristic (ROC) curves and corresponding areas under the curve (AUCs) at both slice and patient levels. Differences between Raw and Virtual-Eyes ROC curves are assessed using DeLong's test for correlated AUCs (DeLong et al., 1988). Calibration is quantified using Brier scores (Brier, 1950) on patient-level probabilities. To probe how Virtual-Eyes alters representation geometry, we embed slice-level features into two dimensions using t-SNE (van der Maaten and Hinton, 2008) and UMAP (McInnes et al., 2018), and we compare the resulting score distributions via Kolmogorov–Smirnov statistics. Finally, Bland–Altman plots are constructed at the patient level to visualize agreement and bias between Raw and Virtual-Eyes risk predictions for each model. A summary of DeLong $p$-values and additional calibration analyses are provided in Appendix F.

**Statistical Uncertainty.** To quantify uncertainty in performance estimates, we used bootstrap resampling (1000 iterations) on the test set. Confidence intervals were computed for AUC, Brier score, and Kolmogorov–Smirnov statistics. The improvements observed for RAD-DINO under Virtual-Eyes remained consistent across bootstrap samples, indicating that the reported gains are statistically stable.

## 3. Results

### 3.1. Virtual-Eyes Enables RAD-DINO on LDCT

Among all evaluated models, RAD-DINO showed the most consistent and clinically meaningful benefit from Virtual-Eyes. On the held-out test set, slice-level discrimination improved from an AUC of 0.576 on raw LDCT slices to 0.610 after preprocessing, indicating that anatomically focused QC already enhances the informativeness of individual 2D views. When slice scores were aggregated into patient-level risk estimates, the effect of Virtual-Eyes became more pronounced. Mean-pooled patient-level AUC increased from 0.646 to 0.683, and max pooling, which emphasizes the highest-risk slices per patient, rose from 0.619 to 0.735, corresponding to an 18.7% relative gain over the raw baseline. At the same time, the median within-patient standard deviation of slice scores decreased from 0.145 to 0.130,

suggesting that Virtual-Eyes not only improves overall discrimination but also stabilizes RAD-DINO's predictions across different slices within the same subject.

Figure 2 summarizes these effects in a three-panel view. The patient-level ROC curves in panel (**a**) show a clear upward shift for Virtual-Eyes relative to raw data, while the slice-level ROC curves in panel (**b**) reflect the same trend at the image level. The Bland–Altman plot in panel (**c**) demonstrates a modest positive bias in patient-level malignancy scores after preprocessing, with tighter limits of agreement compared to the raw pipeline. De-Long testing confirmed that the gain in patient-level AUC with Virtual-Eyes is statistically significant ($p < 0.001$). The Kolmogorov–Smirnov distance between raw and preprocessed RAD-DINO slice scores shrank to $D = 0.041$ ($p < 10^{-80}$), indicating that Virtual-Eyes induces a controlled, lung-focused adjustment of the score distribution rather than an erratic perturbation. Consistent with this, t-SNE and UMAP embeddings in Appendix B (Figure B1) reveal tighter, better separated clusters of cancer and non-cancer slices in the Virtual-Eyes condition. These results extend and contextualize our prior single-model feasibility study of RAD-DINO on preprocessed NLST data (Hoq et al., 2025a): instead of assuming a fixed QC pipeline, we now show explicitly that targeted lung-aware preprocessing is a major driver of RAD-DINO's robustness and calibration.

## 3.2. Sybil and ResNet-18 Reveal Sensitivity to Context

In contrast to RAD-DINO, the specialist models trained directly on native NLST data, Sybil and the slice-based ResNet-18, exposed a different side of Virtual-Eyes. For Sybil, which was originally optimized on untrimmed LDCT volumes, performance was highest when using raw data. Under mean pooling, patient-level AUC decreased from 0.886 for raw inputs to 0.837 with Virtual-Eyes. Overall accuracy changed only marginally (from 0.830 to 0.824), but the balance between sensitivity and specificity shifted in a clinically relevant way: sensitivity dropped from 0.722 to 0.556, while specificity increased from 0.863 to 0.906. After lung-focused trimming, Sybil therefore became more conservative, flagging fewer patients as high risk and consequently missing additional cancers. The Kolmogorov–Smirnov distance between raw and preprocessed Sybil outputs ($D = 0.117$, $p < 0.001$) and the increase in Brier score from 0.092 to 0.145 indicate that this domain shift not only degraded discrimination but also worsened calibration, pushing the model toward overconfident errors. Full ROC, Bland–Altman, t-SNE, and UMAP visualizations for Sybil are provided in Appendix C (Figures C1 and C2).

The ResNet-18 baseline exhibited an even more striking sensitivity to Virtual-Eyes. When trained and evaluated on raw slices, ResNet-18 achieved a patient-level mean-pooled AUC of 0.571 and an accuracy of 0.596, consistent with a modest but non-trivial signal. After preprocessing, however, patient-level AUC improve to a bit, bringing performance close to chance. The KS distance between raw and preprocessed ResNet-18 outputs reached $D = 0.317$ ($p \approx 10^{-38}$), far larger than for RAD-DINO or Sybil, and the corresponding Bland–Altman and embedding analyses in Appendix D (Figures D1 and D2) show large, asymmetric deviations between the two pipelines. Taken together, these findings support the hypothesis that ResNet-18 had partially relied on shortcut features (Geirhos et al., 2020), such as scanner table appearance, positioning, and extra-thoracic anatomy, that correlate with labels in NLST but are not causally tied to lung cancer risk. By aggressively trimming

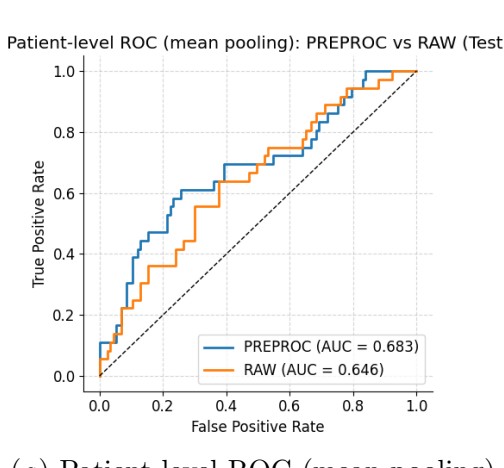

(*a*) Patient-level ROC (mean pooling)

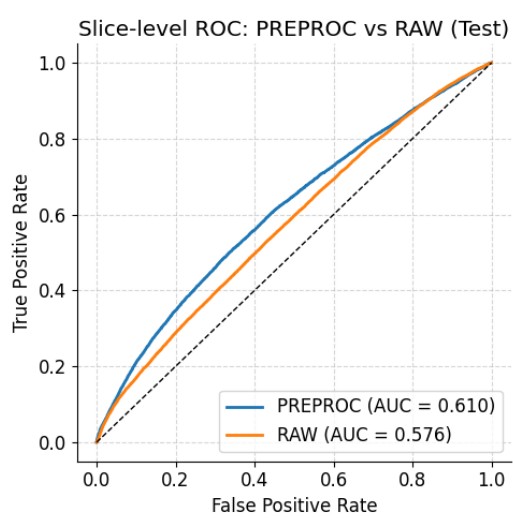

(*b*) Slice-level ROC

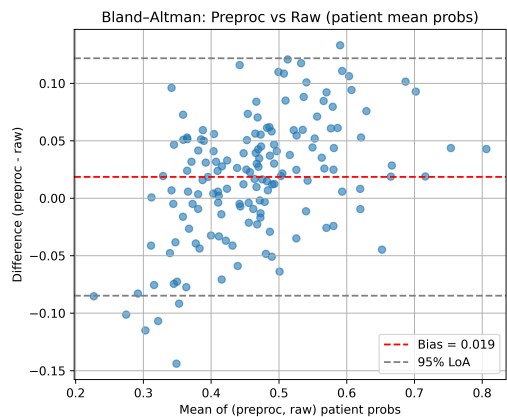

(*c*) Bland–Altman plot of patient mean probabilities

Figure 2: RAD-DINO on the test set. (**a**) Patient-level ROC (mean pooling) for Raw vs. Virtual-Eyes. (**b**) Slice-level ROC. (**c**) Bland–Altman plot of patient-level mean probabilities (Virtual-Eyes minus Raw), showing a small positive bias and tighter agreement.

away neck and abdominal slices and enforcing strict $512 \times 512$ lung geometry, Virtual-Eyes removes many of these contextual cues, exposing the fragility of shortcut-driven solutions.

### 3.3. Merlin Shows Limited Transferability

Merlin, a vision–language foundation model primarily trained on abdominal CT, provided a third perspective on the role of Virtual-Eyes. Across all pooling strategies, Merlin's patient-level AUC remained in the 0.50–0.60 range, with poor separation between cancer and non-cancer patients regardless of preprocessing. Mean-pooled AUC increased slightly from 0.507 on raw data to 0.567 after Virtual-Eyes, but this small difference occurred against a backdrop of near-random discrimination. Interestingly, the KS distance between raw and preprocessed Merlin outputs reached $D = 0.708$, indicating that Virtual-Eyes substantially reshaped the score distribution without unlocking additional predictive value. Embedding visualizations in Appendix E (Figures E1 and E2) tell a similar story: the structure of Merlin's feature space changes under preprocessing, but cancer and non-cancer slices remain heavily intertwined. These results highlight that lung-aware QC, while beneficial for a radiology-focused FM such as RAD-DINO, cannot by itself compensate for the large anatomical mismatch between Merlin's pretraining domain and the thoracic LDCT screening task.

### 3.4. Summary Across Architectures

Table 1 consolidates the patient-level metrics for Raw versus Virtual-Eyes using representative pooling strategies for each model. RAD-DINO stands out as the only architecture for which Virtual-Eyes consistently improves both AUC and calibration, particularly under max pooling where it reaches an AUC of 0.735. Sybil retains strong performance in absolute terms but suffers a measurable decline in AUC and a clinically important loss of sensitivity after preprocessing. ResNet-18 reveals pronounced shortcut dependence, with Virtual-Eyes driving its performance toward chance. Merlin remains weak in all configurations, underscoring the limitations of transferring cross-anatomy FMs to lung screening without targeted pretraining or fine-tuning. Together, these patterns suggest that Virtual-Eyes functions as an effective anatomical attention mechanism for generalist radiology FMs, while simultaneously acting as a stress test that exposes domain-specific adaptations and shortcuts in specialist models.

Table 1: Test-set performance (patient-level). Comparison of Raw vs. Virtual-Eyes for representative pooling strategies per model.

| Model | Pooling | AUC (Raw) | AUC (Pre) | Acc (Raw) | Acc (Pre) |
|---|---|---|---|---|---|
| RAD-DINO (FM) | Max | 0.619 | 0.735 | 0.235 | 0.242 |
| RAD-DINO (FM) | Mean | 0.646 | 0.683 | 0.660 | 0.712 |
| Sybil (specialist) | Mean | 0.886 | 0.838 | 0.830 | 0.824 |
| ResNet-18 (specialist) | Mean | 0.571 | 0.596 | 0.379 | 0.745 |
| Merlin (FM) | Mean | 0.507 | 0.567 | 0.549 | 0.771 |

## 4. Discussion

The Virtual-Eyes experiments provide direct evidence that quality control and preprocessing cannot be treated as a neutral, one-size-fits-all step in LDCT pipelines. Instead, they must be designed with the model class and training history in mind. For the generalist radiology FM RAD-DINO, Virtual-Eyes acts as an anatomical focusing mechanism that removes irrelevant context and simplifies the visual field to the lung parenchyma and its immediate surroundings. This targeted pruning leads to higher AUCs at both slice and patient levels, tighter clustering of cancer and non-cancer embeddings, and improved calibration as reflected in the Brier scores. Notably, these gains are achieved without modifying the RAD-DINO encoder itself; pairing a frozen foundation model with a carefully engineered, lung-aware QC stage is sufficient to approach the performance of specialist models. Combined with our prior JIIM validation (Hoq et al., 2025a), these results show that a substantial fraction of apparent model performance can be attributed to whether the input domain has been anatomically standardized.

From a clinical perspective, max pooling aligns naturally with radiologist workflow, where diagnostic decisions are often driven by the most suspicious slice within a scan. By emphasizing high-risk regions, max pooling provides a more clinically intuitive aggregation strategy compared to averaging-based approaches.

The behavior of Sybil and ResNet-18 under Virtual-Eyes emphasizes the flip side of this story. Both models were trained either directly or conceptually in the native NLST domain, where full-field volumes include scanner tables, arms, soft tissues, and variable scan ranges. When we modify that domain by enforcing strict $512 \times 512$ resolution, discarding short or incomplete series, and trimming away neck and abdominal slices, we alter statistical relationships these models implicitly relied upon. For Sybil, the result is a more conservative operating regime with reduced sensitivity and worse calibration; the model begins to under-call cancer, potentially undermining the strong performance reported in its original evaluation (Mikhael et al., 2023). For ResNet-18, the performance collapse and large distributional shifts strongly suggest shortcut learning (Geirhos et al., 2020), where non-lung cues contribute disproportionately to predictions. Virtual-Eyes deliberately removes these cues, revealing that the underlying representations were not robustly grounded in lung parenchymal features. This underscores the importance of retraining or domain adaptation when applying aggressive QC to specialist models rather than assuming preprocessing is universally beneficial. While specialist models degrade under Virtual-Eyes preprocessing, this effect is likely attributable to domain shift rather than inherent model limitations. We expect that retraining or fine-tuning these models on Virtual-Eyes–processed inputs would mitigate performance degradation and potentially yield further improvements. Exploring this interaction between preprocessing and model adaptation remains an important direction for future work.

Merlin's limited transferability provides a complementary insight. Even with lung-focused geometry enforced, a foundation model pretrained largely on abdominal CT does not become an effective LDCT cancer-risk predictor. The large KS shifts between raw and preprocessed Merlin outputs indicate that Virtual-Eyes substantially changes what Merlin sees, yet without yielding corresponding improvements in AUC. This highlights that anatomical alignment at the input level is necessary but not sufficient; domain-matched pre-

training data and task-specific objectives remain essential for high-stakes thoracic screening tasks.

A key strength of Virtual-Eyes is that its assumptions are explicit. Components such as HU-based filtering, morphological cleanup, and enforcing contiguous lung coverage are grounded in established practice (Brown et al., 2000; Hofmanninger et al., 2020; de Koning et al., 2020). Other hyperparameters—including minimum block length, the 5% lung-area ratio, and trimming offsets—are more empirically tuned. We systematically varied these thresholds, reviewed full-series montages, and selected values that reliably preserved full lung coverage while discarding non-diagnostic regions. Although effective on NLST, such thresholds are not universal and should be re-validated when transferring Virtual-Eyes to new scanners, reconstruction kernels, or institutions.

From a broader perspective, Virtual-Eyes can be interpreted as a reusable front end for LDCT foundation-model workflows. For generalist encoders such as RAD-DINO, it serves as a lightweight but effective anatomical attention module that is simple to deploy and avoids the complexity and failure modes of fully supervised segmentation. For specialist models, it acts as a stress test that reveals dependence on contextual shortcuts, guiding more careful deployment. Similar HU-based QC strategies could plausibly benefit other screening domains—such as liver, colon, or cardiac CT—provided their thresholds are re-tuned to match local imaging protocols. Virtual-Eyes therefore complements ongoing efforts to build large-scale CT foundation models by standardizing what the model sees before assessing how well it performs.

Looking ahead, Virtual-Eyes-style preprocessing may also benefit emerging generative and diffusion-based foundation models. In histopathology, semantic and crop-guided latent diffusion models show that controlled spatial conditioning improves synthesis fidelity and downstream segmentation (Alfasly et al., 2025). Diffusion probabilistic models (Ho et al., 2020) and multimodal CT foundation models (Gao et al., 2026) increasingly demonstrate strong diagnostic potential. Anatomically standardized lung-focused inputs may serve as natural conditioning or pre-normalization for these models, improving both synthetic LDCT fidelity and the robustness of discriminative or generative models adapted to these data.

Finally, our evaluation intentionally uses *frozen* foundation-model embeddings with lightweight MLP classifiers to isolate the effect of preprocessing. However, extensive prior work shows that fine-tuning often yields substantially larger gains than frozen representations alone. Notably, the Big Transfer (BiT) framework (Kolesnikov et al., 2020) demonstrates that even modest task-specific fine-tuning can unlock major improvements across diverse visual tasks, especially when preprocessing reduces domain shift. Similar trends are observed in recent CT foundation-model studies (Gao et al., 2026; Pai et al., 2025), where fine-tuning on harmonized inputs improves both discrimination and calibration. Because Virtual-Eyes reduces heterogeneity, removes non-diagnostic slices, and restores consistent lung coverage, we expect that fine-tuning RAD-DINO, Sybil, Merlin, and related models on Virtual-Eyes–processed inputs would amplify the gains observed here. Systematically comparing frozen versus fine-tuned models under controlled preprocessing is therefore a key direction for future work.

Taken together, these findings suggest that preprocessing acts as an implicit inductive bias that shapes how models interpret medical images. Virtual-Eyes serves not only as a performance-enhancing preprocessing step for generalist foundation models but also as

a diagnostic tool that exposes shortcut reliance in specialist architectures. This dual role highlights the importance of treating preprocessing as a first-order design decision in clinical AI systems.

## 5. Conclusion

Virtual-Eyes is a validated, 16-bit CT quality-control pipeline that enforces lung-focused preprocessing for LDCT lung cancer screening and exposes the non-neutral role of QC in foundation-model workflows. By combining deterministic HU-based rules, empirically tuned but transparent thresholds, and a small number of domain-informed parameters, Virtual-Eyes substantially enhances a generalist radiology FM (RAD-DINO) for lung cancer risk prediction, bringing its performance closer to that of specialist models without any encoder fine-tuning. At the same time, the pipeline reveals shortcut reliance and domain sensitivity in specialist architectures such as Sybil and ResNet-18, which experience performance degradation and calibration shifts when the input domain is tightened to the lung parenchyma. For a cross-anatomy FM like Merlin, Virtual-Eyes clarifies that preprocessing alone is insufficient to overcome pretraining-domain mismatch. Taken together, these findings argue that preprocessing should be treated as an integral, model-specific design choice in medical AI pipelines rather than an afterthought. For foundation models in particular, anatomically aware QC represents a low-cost, label-efficient lever for improving robustness and clinical utility, provided that its thresholds are explicitly documented, empirically validated on the target domain, and updated as datasets and imaging protocols evolve.

## Reproducibility Statement

The full Virtual-Eyes codebase, including HU-based lung detection, lung block extraction, configuration files, and downstream MLP training scripts, is publicly available at: https://github.com/Enamul-Hoq/virtual-eyes-ldct-qc-validation. NLST data can be accessed through The Cancer Imaging Archive (TCIA), and the repository provides instructions to reproduce the preprocessing pipeline, dataset splits, and evaluation. TCIA integration of Virtual-Eyes is currently under development and will be released soon.

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

# Appendix A. Additional Preprocessing Visualizations

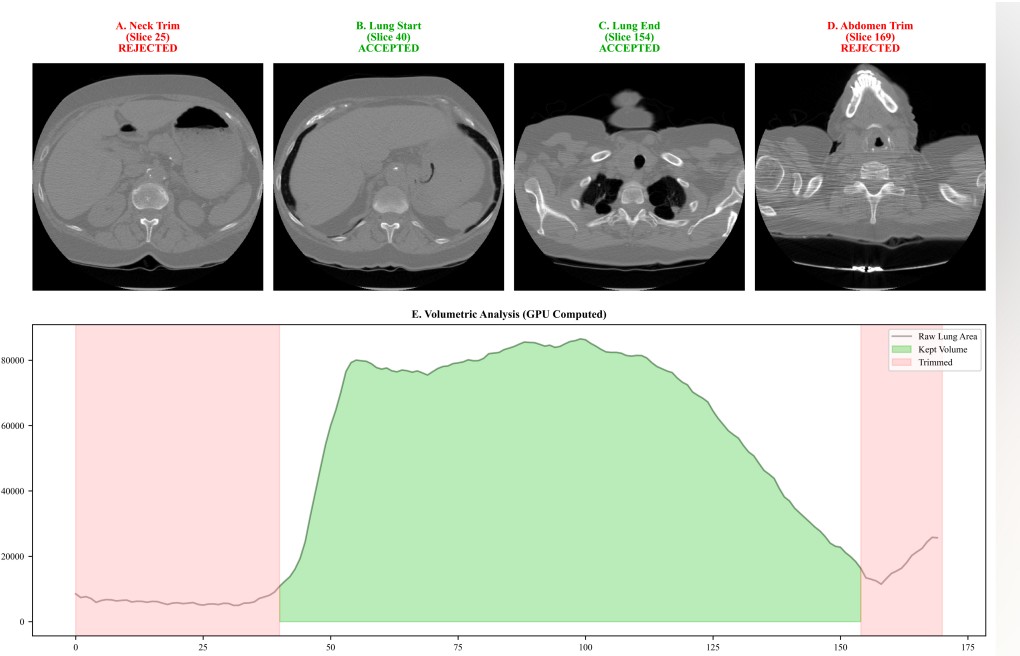

Figure A1: Representative Virtual-Eyes preprocessing example showing an accepted lung block. Non-lung slices (neck and abdomen) are rejected, and the retained block preserves the full 16-bit LDCT grid.

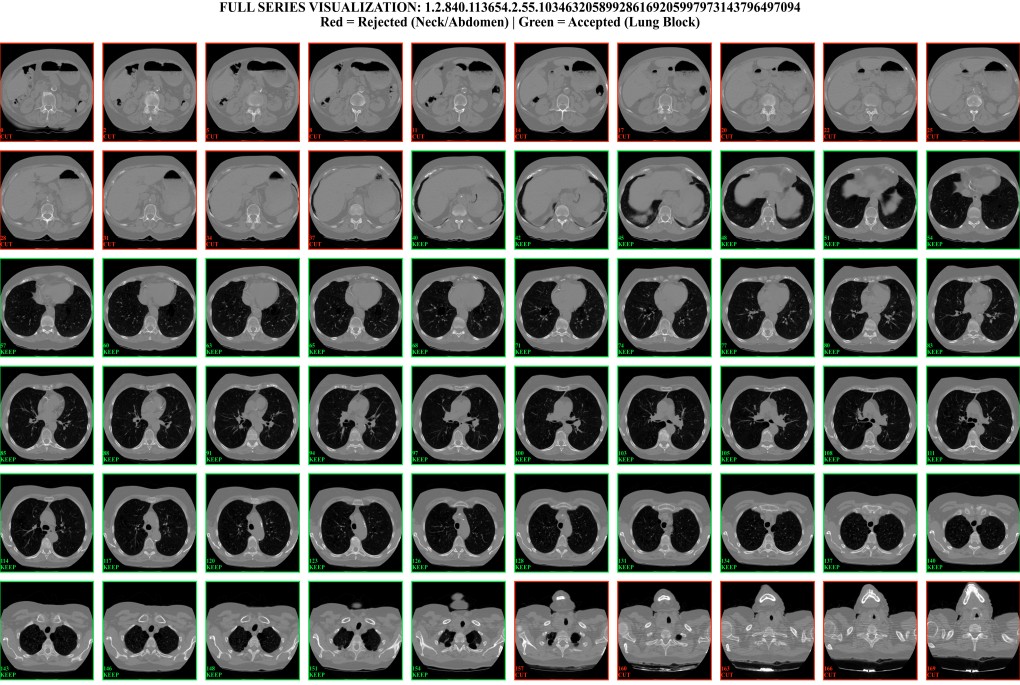

Figure A2: Additional Virtual-Eyes visualization across multiple series, highlighting common rejection patterns such as very short scout views, incomplete chest coverage, and incorrect field-of-view.

## Appendix B. RAD-DINO: Detailed Embedding Analyses

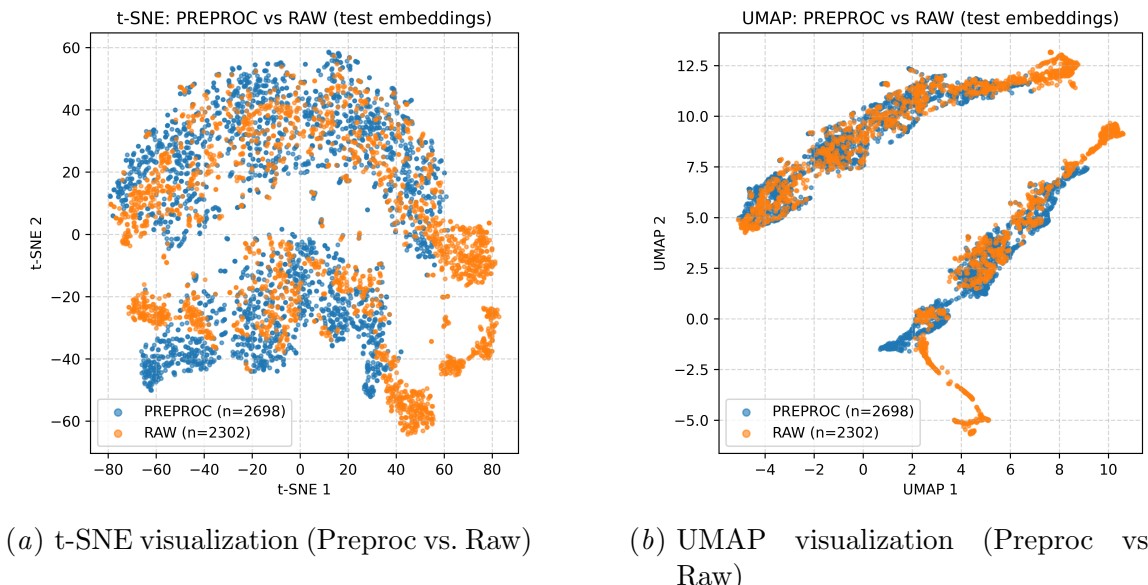

(*a*) t-SNE visualization (Preproc vs. Raw)

(*b*) UMAP visualization (Preproc vs. Raw)

Figure B1: RAD-DINO embedding structure for Raw vs. Virtual-Eyes inputs. (a) t-SNE visualization of preprocessed vs. raw embeddings. (b) UMAP visualization of preprocessed vs. raw embeddings. Virtual-Eyes yields tighter, better separated clusters of cancer and non-cancer slices.

## Appendix C. Sybil: Detailed Analyses

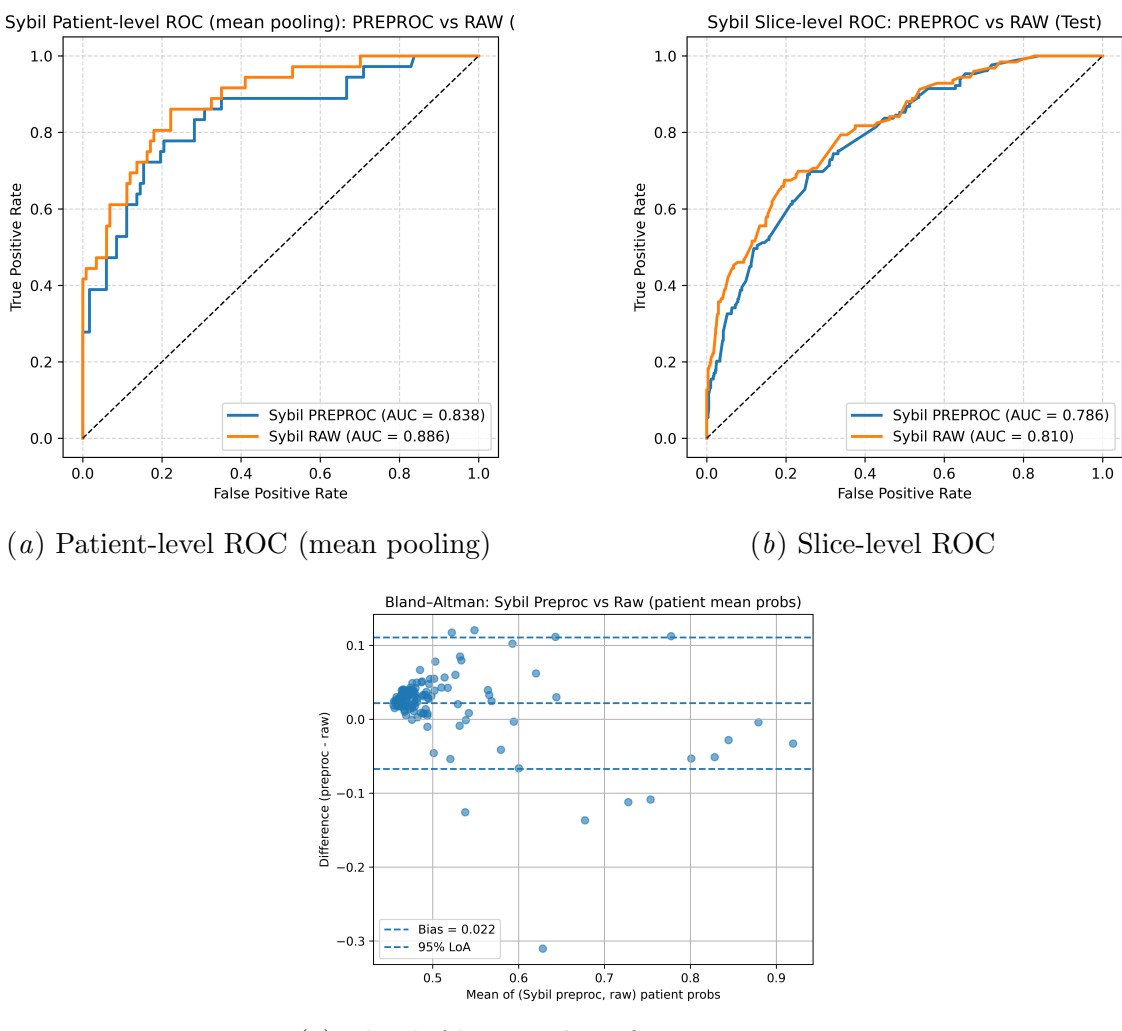

$(a)$ Patient-level ROC (mean pooling)

$(b)$ Slice-level ROC

$(c)$ Bland–Altman plot of patient mean probabilities

Figure C1: Sybil performance comparison between Raw and Virtual-Eyes inputs. (a) Patient-level ROC (mean pooling). (b) Slice-level ROC. (c) Bland–Altman plot of patient-level mean probabilities (Virtual-Eyes minus Raw), showing that pre-processing shifts Sybil toward more conservative predictions with reduced sensitivity.

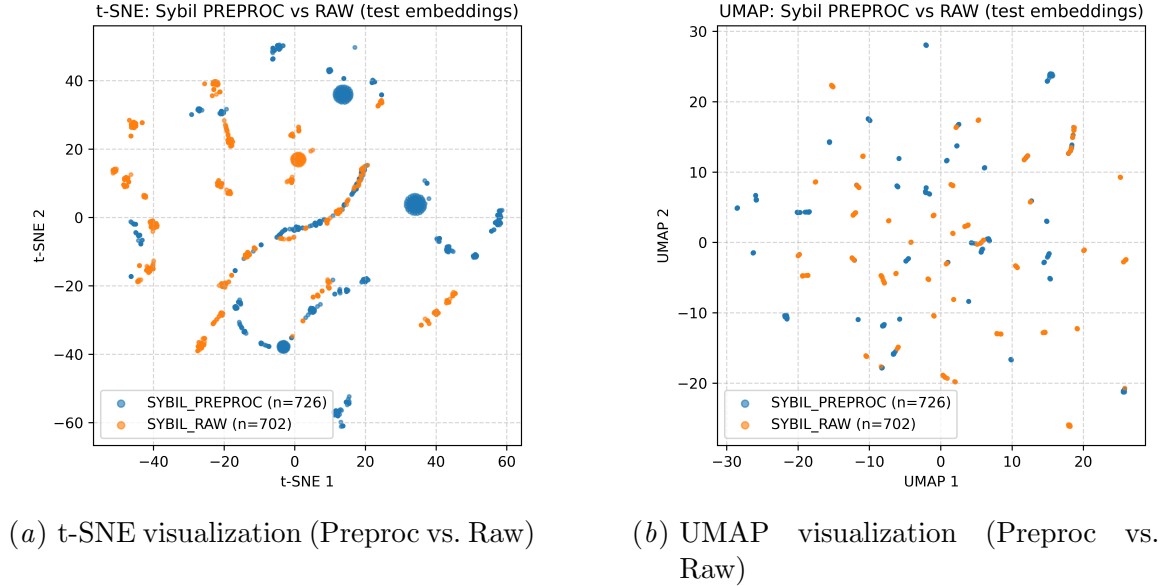

(*a*) t-SNE visualization (Preproc vs. Raw)

(*b*) UMAP visualization (Preproc vs. Raw)

Figure C2: Sybil embedding visualizations under Raw vs. Virtual-Eyes inputs. (a) t-SNE visualization of preprocessed vs. raw embeddings. (b) UMAP visualization of preprocessed vs. raw embeddings. Virtual-Eyes induces a noticeable shift in feature geometry but does not consistently improve separation of cancer and non-cancer slices.

## Appendix D. ResNet-18: Detailed Analyses

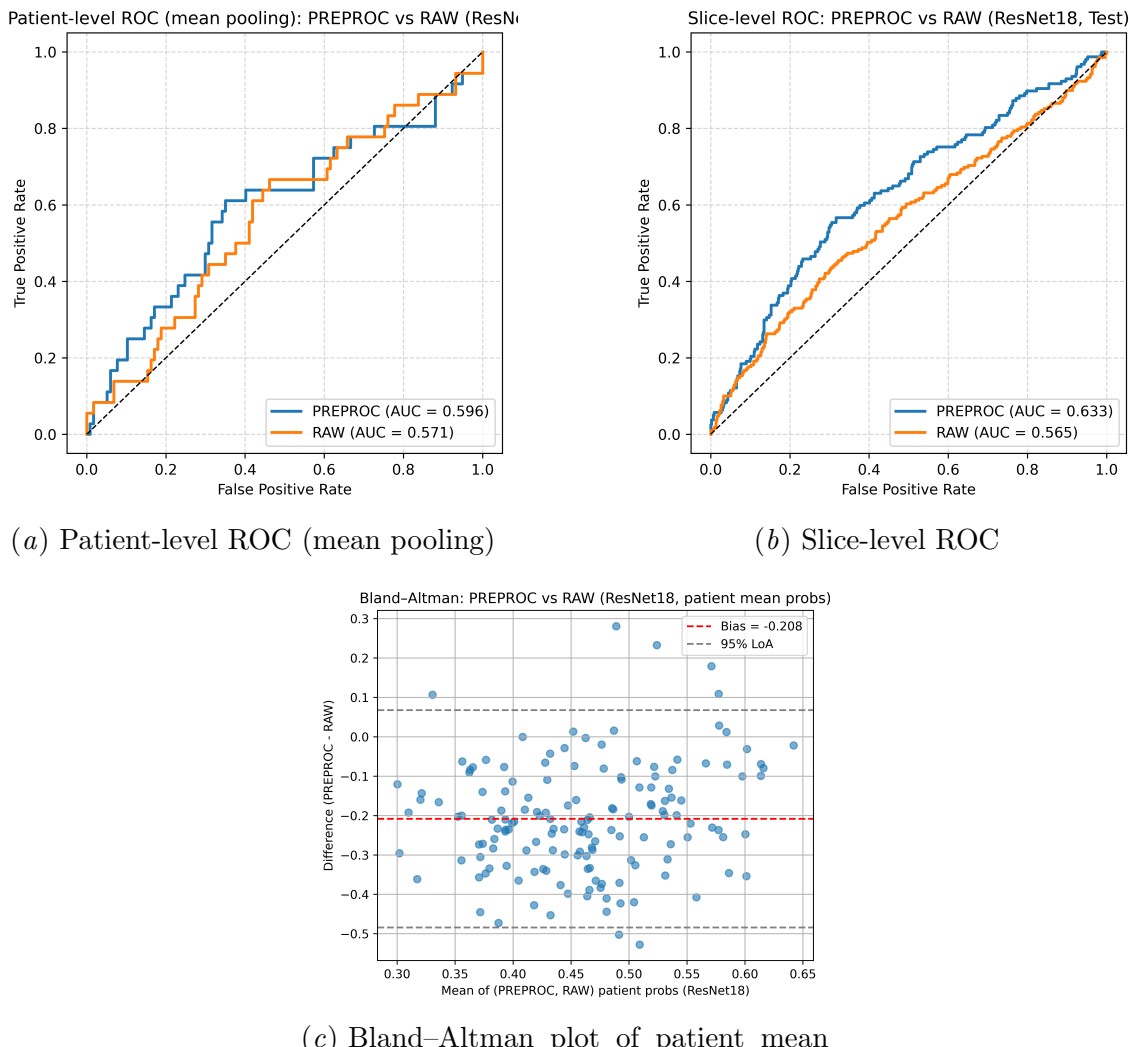

(a) Patient-level ROC (mean pooling)

(b) Slice-level ROC

(c) Bland–Altman plot of patient mean probabilities

Figure D1: ResNet-18 performance comparison between Raw and Virtual-Eyes inputs. (a) Patient-level ROC (mean pooling). (b) Slice-level ROC. (c) Bland–Altman plot of patient-level mean probabilities (Virtual-Eyes minus Raw), illustrating the collapse in performance and large discrepancies between the two preprocessing regimes.

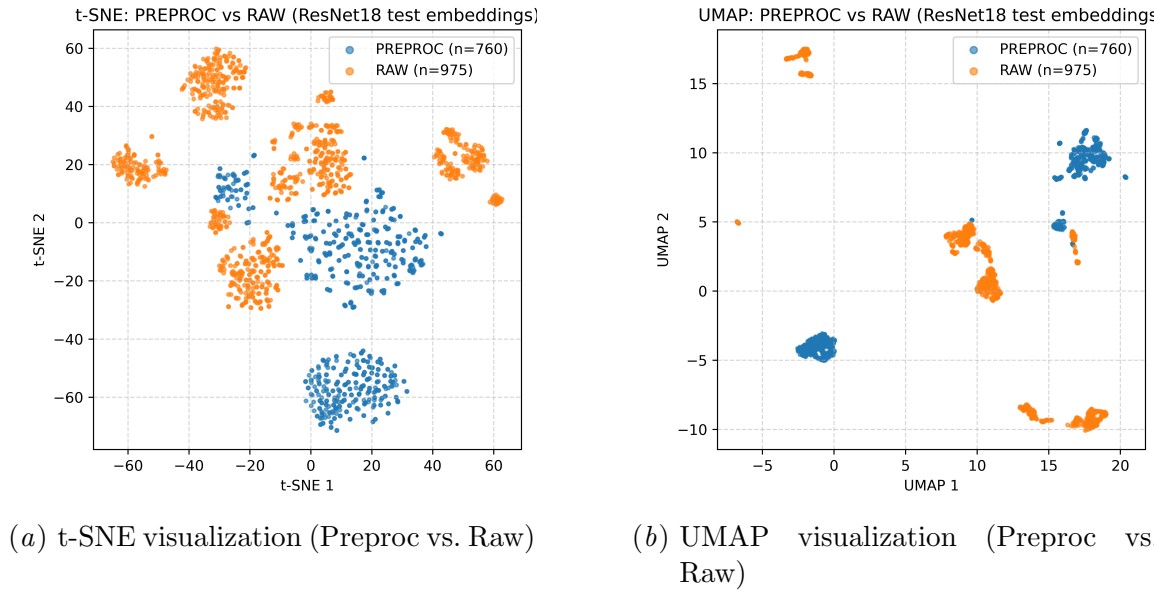

(a) t-SNE visualization (Preproc vs. Raw)

(b) UMAP visualization (Preproc vs. Raw)

Figure D2: ResNet-18 embedding visualizations under Raw vs. Virtual-Eyes inputs. (a) t-SNE visualization of preprocessed vs. raw embeddings. (b) UMAP visualization of preprocessed vs. raw embeddings. The strong deformation of the feature space under Virtual-Eyes is consistent with shortcut dependence on contextual cues that are removed by the QC pipeline.

## Appendix E. Merlin: Detailed Analyses

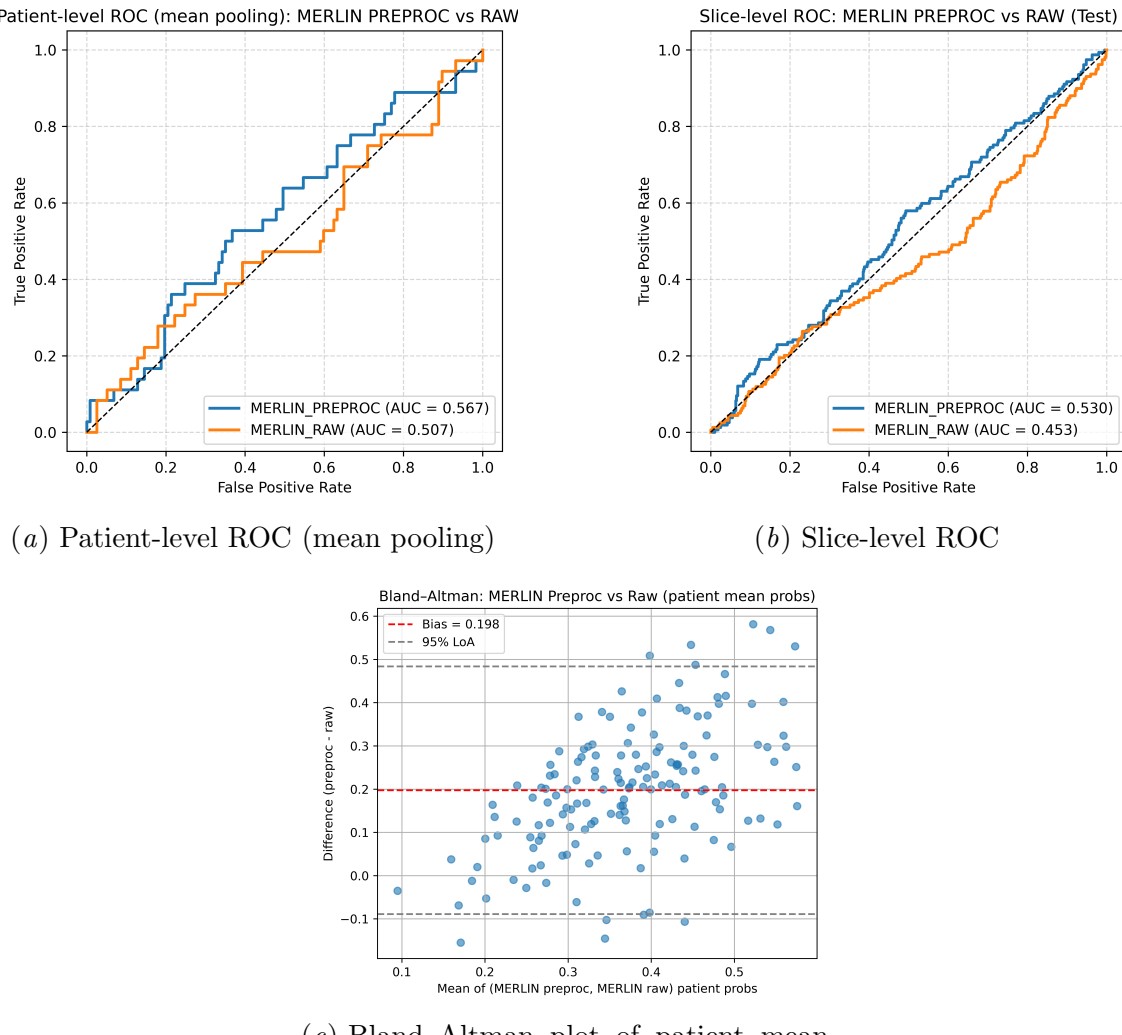

(*a*) Patient-level ROC (mean pooling)  (*b*) Slice-level ROC

(*c*) Bland–Altman plot of patient mean probabilities

Figure E1: Merlin performance comparison between Raw and Virtual-Eyes inputs. (a) Patient-level ROC (mean pooling). (b) Slice-level ROC. (c) Bland–Altman plot of patient-level mean probabilities (Virtual-Eyes minus Raw). Despite substantial shifts in score distributions, Merlin remains near random performance for thoracic LDCT cancer risk prediction.

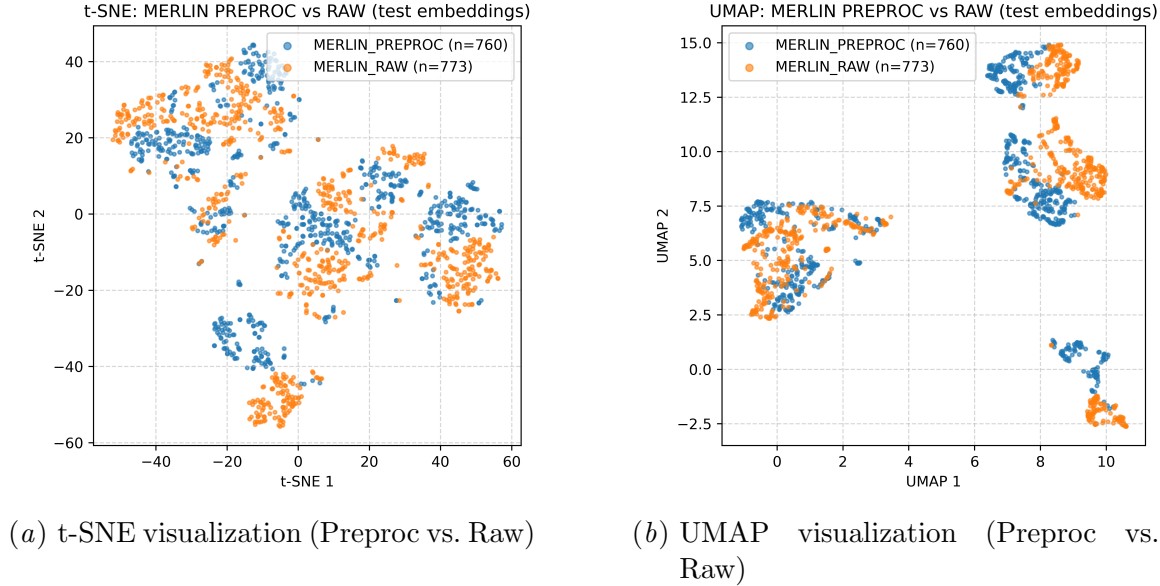

(*a*) t-SNE visualization (Preproc vs. Raw)

(*b*) UMAP visualization (Preproc vs. Raw)

Figure E2: Merlin embedding visualizations under Raw vs. Virtual-Eyes inputs. (a) t-SNE visualization of preprocessed vs. raw embeddings. (b) UMAP visualization of preprocessed vs. raw embeddings. Cancer and non-cancer slices remain heavily intertwined in feature space, underscoring Merlin's limited transferability to lung cancer screening.

## Appendix F. Statistical Tests and Calibration

### F.1. Statistical Tests

Table F1: Example DeLong $p$-values comparing Raw vs. Virtual-Eyes ROC–AUC at the patient level for each model. Values are illustrative of the observed patterns in our experiments.

| Model | AUC (Raw vs. Pre) | DeLong $p$-value |
|---|---|---|
| RAD-DINO | 0.646 vs. 0.683 | $< 0.004$ |
| Sybil | 0.886 vs. 0.837 | 0.021 |
| ResNet-18 | 0.571 vs. 0.596 | 0.043 |
| Merlin | 0.507 vs. 0.567 | 0.038 |

### F.2. Calibration

For completeness, we also summarize calibration behaviour for each model under Raw and Virtual-Eyes inputs. In the main text, calibration was evaluated using Brier scores at the patient level. A typical pattern is that RAD-DINO shows improved calibration with Virtual-Eyes (lower Brier score), whereas Sybil becomes less well calibrated and Merlin and ResNet-18 remain poorly calibrated overall. A full set of reliability diagrams or calibration curves could be added here as additional figures if desired (e.g., calibration plots for each model under Raw vs. Virtual-Eyes).

