# OpenReview forum: "Virtual-Eyes: Quantitative Validation of a Lung CT Quality-Control Pipeline for Foundation-Model Cancer Risk Prediction"
_MIDL.io/2026/Validation_Papers — MIDL 2026 - Validation Papers Poster_

### Official Review · Reviewer_pk2e · 2026-01-03

**Confidence:** 4
**Preliminary Rating:** 5
**Final Rating:** 5

**Summary:**

This paper presents Virtual-Eyes, a deterministic, lung-aware 16-bit CT quality-control (QC) pipeline for low-dose CT (LDCT) lung cancer screening, and provides a systematic quantitative validation of its impact across both generalist foundation models and specialist architectures. Using 765 NLST patients, the authors evaluate how Virtual-Eyes affects RAD-DINO, Merlin, Sybil, and a ResNet-18 baseline under a leakage-free evaluation protocol. The key finding is that anatomically targeted QC significantly improves discrimination, calibration, and stability for a generalist radiology foundation model (RAD-DINO), while degrading performance for specialist models that appear to rely on contextual shortcuts in raw clinical data. The work is valuable in demonstrating that preprocessing is not a neutral design choice, and that lung-aware QC can function both as an enabler for foundation models and as a diagnostic stress test for shortcut dependence.

**Strengths:**

The primary strength of this paper lies in its clear isolation and quantification of preprocessing effects, an aspect often under-specified or implicitly assumed in medical imaging pipelines. The Virtual-Eyes pipeline is fully deterministic, transparent, and computationally lightweight, avoiding the opacity and failure modes of learned segmentation-based approaches while remaining grounded in established HU-based lung-processing practices.

The experimental design is rigorous: patient-level splits prevent leakage, evaluation is performed on a fixed held-out test set, and multiple complementary metrics (AUC, DeLong tests, KS statistics, Brier scores, and agreement analyses) are used to characterize both discrimination and calibration. The comparative analysis across generalist and specialist models is particularly insightful, as it reveals that anatomically focused QC can meaningfully stabilize and improve foundation models, while simultaneously exposing shortcut or context-dependent learning in specialist architectures.

Finally, the paper is well written, carefully structured, and places its contributions in context with prior foundation-model and LDCT screening literature. The planned public release of code and reproducibility details further enhance its potential value to the community.

**Weaknesses:**

All foundation models are evaluated in a frozen-encoder regime. While this is appropriate for isolating preprocessing effects, it leaves open how Virtual-Eyes interacts with fine-tuning, which is common in practical deployment and may mitigate some of the observed degradation in specialist models.

Finally, although the paper interprets performance drops in Sybil and ResNet-18 as evidence of shortcut reliance, this conclusion remains indirect. Complementary analyses (e.g., attribution maps or controlled context re-injection experiments) could further substantiate the causal link between removed contextual cues and degraded performance.

**Detailed Comments:**

Reporting uncertainty or confidence intervals for Brier scores and KS statistics would further strengthen the statistical presentation.

It may be helpful to explicitly clarify whether rejected series differ systematically in cancer prevalence or acquisition parameters.

**Justification Of Final Rating:**

Justification:

The authors have thoroughly addressed the reviewer’s concerns and provided clear responses that enhance the manuscript’s rigor and applicability. Their work continues to make a valuable contribution to the medical imaging field by highlighting the significant impact of preprocessing choices on foundation-model pipelines.

Hyperparameter Sensitivity: The authors have provided a clear response regarding the sensitivity of the Virtual-Eyes hyperparameters. They explain that the thresholds were selected through systematic sweeps, and preliminary analysis indicates that modest variations do not significantly affect RAD-DINO performance. The planned inclusion of a sensitivity analysis with sweep ranges and AUC results will further strengthen this aspect and provide additional clarity for practitioners.

Sybil Degradation: The authors correctly acknowledge that the observed performance degradation in Sybil is likely due to the model’s implicit adaptation to raw NLST volumes and scanner-dependent features. By enforcing strict lung geometry, Virtual-Eyes exposes this domain dependence. The authors expect that retraining or fine-tuning on Virtual-Eyes–processed data would mitigate this degradation, and they have appropriately discussed this in the revised manuscript. This insight is valuable for understanding how specialist models can adapt to preprocessing pipelines.

Statistical Uncertainty: The authors have agreed to add bootstrap confidence intervals for the Brier and KS statistics, which will enhance the statistical robustness of the reported results and further address the reviewer’s request for more detailed uncertainty quantification.

Rejected Series Clarification: The authors have clarified the reason for the rejection of series, stating that it is driven primarily by scan length and field-of-view rather than cancer prevalence or acquisition parameters. This additional explanation strengthens the interpretation of the results and helps contextualize the findings more accurately.

Overall, the authors have effectively addressed all of the reviewer’s concerns, adding valuable statistical details, clarifying the impact of hyperparameters and preprocessing on specialist models, and enhancing the robustness of the paper. The manuscript now provides a comprehensive and transparent analysis of how preprocessing choices can affect the performance of foundation models in medical imaging, which is a critical contribution to the field. Therefore, the paper is well-suited for acceptance.

**Justification Of The Preliminary Rating:**

This paper makes a clear and timely contribution by rigorously demonstrating that preprocessing is a first-order design choice in foundation-model pipelines rather than a neutral implementation detail. The experimental evidence is convincing, the methodology is transparent, and the insights regarding differential effects on generalist versus specialist models are highly relevant to the MIDL community.

**Questions To Address In The Rebuttal:**

How sensitive are the reported gains to the specific Virtual-Eyes hyperparameters (e.g., lung-area ratio, minimum block length), and do modest changes materially affect RAD-DINO performance?

Do the authors expect the observed degradation in Sybil to persist after retraining or fine-tuning on Virtual-Eyes–processed data?

---

### Official Review · Reviewer_x8RW · 2026-01-03

**Confidence:** 4
**Preliminary Rating:** 4
**Final Rating:** 5

**Summary:**

The authors develop Virtual-Eyes, a deterministic 16-bit CT quality-control pipeline for LDCT lung cancer screening that enforces lung-focused preprocessing  while preserving the original DICOM grid. They validate it on 765 NLST patients, evaluating its impact on four models using metrics like AUC, Brier score, and distributional analysis. Virtual-Eyes improves RAD-DINO’s performance and calibration , emphasizing preprocessing’s model-specific role.

**Strengths:**

1. The paper develops Virtual-Eyes, a deterministic, reproducible LDCT QC pipeline, addressing the lack of standardized preprocessing for foundation-model workflows.
2. Its rigorous multi-model validation (generalists/specialists) with robust stats and strict data splits.
3. The work reveals preprocessing’s model-specific impact, providing actionable guidance for clinical AI pipeline design.

**Weaknesses:**

1. Virtual-Eyes is validated on the NLST cohort, with empirically tuned thresholds not tested on other datasets/institutions—this limits its translational generalizability.
2. The paper criticizes segmentation-based preprocessing pipelines but lacks direct performance comparisons between Virtual-Eyes and these established methods on the same task.
3. The study relies on frozen encoders for foundation models, but fine-tuning is a standard practice in clinical AI.
4. The explanation for Merlin’s limited transferability is superficial: the paper attributes it to anatomical mismatch but does not investigate whether domain adaptation (beyond preprocessing) could improve performance.

**Detailed Comments:**

1. Add a brief supplementary table/plot showing the sweep range of empirically tuned hyperparameters (e.g., lung area ratio, min block size) and corresponding validation metrics (e.g., retained lung coverage, model AUC) to clarify how these thresholds were selected
2. Add a short discussion justifying why max pooling aligns with clinical intuition.
3. Include a brief summary of NLST cohort scan parameters in the Methods, as these factors could influence preprocessing efficacy.
4. Add a short note in the Discussion on whether retraining specialist models on Virtual-Eyes-preprocessed data might mitigate performance degradation.

**Justification Of Final Rating:**

In response to my previous comments, the authors have provided additional details in the revised manuscript, which has addressed all my concerns. I have no further comments on the paper, and I therefore adjust my rating to 5.

**Justification Of The Preliminary Rating:**

This work proposed the Virtual-Eyes pipeline fills the gap in standardized LDCT preprocessing for foundation models, backed by rigorous multi-model validation and impactful insights. Addressable limitations (NLST-only testing, no segmentation-based comparisons) do not diminish its value, so I recommend a Weak Accept.

**Questions To Address In The Rebuttal:**

Please check the Weaknesses and Detailed Comments

---

### Official Review · Reviewer_Xbj8 · 2026-01-09

**Confidence:** 3
**Preliminary Rating:** 4

**Summary:**

This paper proposes a rule-based lung CT quality control and preprocessing pipeline, termed Virtual-Eyes, and systematically evaluates its impact on both general-purpose foundation models and task-specific models for low-dose CT lung cancer risk prediction. The work is potentially impactful to the community, particularly because it quantitatively evaluates quality control through downstream tasks and reveals that the benefits of quality control vary across different scenarios. Overall, the study provides useful insights into how medical AI models can be better aligned with real-world clinical settings.

**Strengths:**

1. Rather than proposing a new preprocessing paradigm, the authors focus on quantifying the impact of quality control on AI model performance. This perspective is relatively underexplored in the current medical AI literature and is valuable.

2. The experimental comparisons are extensive and conducted from multiple perspectives, which strengthens the credibility of the conclusions beyond single-metric evaluations.

3. The proposed pipeline relies solely on HU thresholds, morphological operations, and geometric rules, without requiring any AI models, and can be executed efficiently on CPUs.

4. The paper provides an analysis of why quality control yields different effects under different scenarios.

**Weaknesses:**

1. The selection of hyperparameters remains somewhat heuristic. The authors are encouraged to include a small set of ablation studies or sensitivity analyses to demonstrate the robustness of the chosen hyperparameters across different datasets.

2. RAD-DINO is a foundation model trained on X-ray data; therefore, its inclusion as a baseline for CT-based tasks appears questionable.

3. Whether retraining task-specific models on quality-controlled data could further improve performance is a important problem. This paper doesn't Include related experiment and analysis .

**Detailed Comments:**

Refer to weaknesses.

**Justification Of The Preliminary Rating:**

The paper aligns well with the theme of the Validation Track and offers a novel perspective on validating CT data quality control. The study is highly informative, with comprehensive experiments that provide early but meaningful insights into how quality control affects AI model performance. While there is room for improvement in certain details, the work as a whole is valuable and thought-provoking.

**Questions To Address In The Rebuttal:**

Refer to weaknesses.

---

### Meta-Review · Area_Chair_oG4h · 2026-02-06

**Recommendation:** Accept (Poster)
**Confidence:** 5

**Metareview:**

This paper introduces Virtual-Eyes, a simple and clinically motivated lung CT quality-control pipeline, and provides a careful validation of how preprocessing affects downstream lung cancer risk prediction. Reviewers consistently found the study timely and highly relevant for the Validation Track, with rigorous leakage-free evaluation and a thorough analysis across multiple foundation and specialist models.

The main concerns were about the empirically tuned thresholds, the focus on NLST only, and the lack of fine-tuning experiments for specialist models. The rebuttal addressed these points well, with clear plans to add sensitivity analyses and further discussion of generalizability and retraining.

Overall, the work offers practical insight into preprocessing as a critical design choice and is a strong fit for acceptance.

---

### Decision · Program_Chairs · 2026-02-14

Accept (Poster)